# Intracerebral Hemorrhage Genetics

**DOI:** 10.3390/genes13071250

**Published:** 2022-07-15

**Authors:** Aleksandra Ekkert, Aleksandra Šliachtenko, Algirdas Utkus, Dalius Jatužis

**Affiliations:** 1Center of Neurology, Faculty of Medicine, Vilnius University, LT-03101 Vilnius, Lithuania; dalius.jatuzis@mf.vu.lt; 2Faculty of Medicine, Vilnius University, LT-03101 Vilnius, Lithuania; aleksandra.sliachtenko@mf.stud.vu.lt; 3Center for Medical Genetics, Faculty of Medicine, Vilnius University, LT-03101 Vilnius, Lithuania; algirdas.utkus@mf.vu.lt

**Keywords:** hemorrhagic stroke genetics, intracerebral hemorrhage, single nucleotide polymorphism, genetic stroke causes

## Abstract

Intracerebral hemorrhage (ICH) is a devastating type of stroke, frequently resulting in unfavorable functional outcomes. Up to 15% of stroke patients experience ICH and approximately half of those have a lethal outcome within a year. Considering the huge burden of ICH, timely prevention and optimized treatment strategies are particularly relevant. Nevertheless, ICH management options are quite limited, despite thorough research. More and more trials highlight the importance of the genetic component in the pathogenesis of ICH. Apart from distinct monogenic disorders of familial character, mostly occurring in younger subjects, there are numerous polygenic risk factors, such as hypertension, neurovascular inflammation, disorders of lipid metabolism and coagulation cascade, and small vessel disease. In this paper we describe gene-related ICH types and underlying mechanisms. We also briefly discuss the emerging treatment options and possible clinical relevance of the genetic findings in ICH management. Although existing data seems of more theoretical and scientific value so far, a growing body of evidence, combined with rapidly evolving experimental research, will probably serve clinicians in the future.

## 1. Introduction

Although intracerebral hemorrhage (ICH) is not the most common form of stroke (responsible for 10–15% of all strokes), it places a great burden on society [1]. It is a significant cause of mortality and disability worldwide with only 50% of sufferers having 1-year survivability [2]. Acute therapies for ICH are still under investigation and their effectiveness is limited, which makes predictive and preventive measures even more important. Unlike ischemic strokes, there are not so many modifiable risk factors in the case of ICH. In this case, hypertension is a major player and the management of high blood pressure was proved to be an effective preventive strategy [3,4]. However, the heritability of ICH is claimed to be up to 44%, and having a first-degree relative with ICH is associated with a six-fold higher risk of ICH, making the genetic component important in the pathogenesis of the disorder [5]. Although the number of potentially modifiable risk factors is modest, investigation of genetic factors and emerging data on gene therapy progress in other fields carry some hope for better individualized ICH prevention and treatment strategies.

ICH is classified according to etiopathogenesis. There are several classification systems based on anatomy or clinics, all of them have proved to be reliable if used properly [6]. The most common “anatomical” classification refers to the region of the brain as lobar (the junction of the cortical gray matter and subcortical white matter), and non-lobar or deep (the thalamus, basal ganglia, brainstem, and cerebellum). The “mechanical” systems use underlying pathological causes, such as hypertension, cerebral amyloid angiopathy (CAA), anticoagulation, and vessels’ structural defects to classify ICH. However, it is hard to use those classification systems separately, as hypertension is often associated with non-lobar ICH, but has little impact on lobar, whereas the most common histopathology found in lobar ICH is cerebral amyloid angiopathy (CAA) [7,8]. The overall heritability of lobar ICH is 48%, 30% for non-lobar ICH, and 29% for both types combined, according to International Stroke Genetics Consortium data [9].

## 2. Methods of Genetic Studies

Most of the information about ICH genetics comes from either the candidate gene or genome-wide association studies (GWAS). Candidate gene selection is based on known risk factors or association with disease. This approach helped to identify several genetic causes of ICH, however, as it relies on prior information, it has serious limitations, such as potential for bias and inability to find completely new genes [10]. On the other hand, GWAS is helpful when one intends to examine the entire genome for new genetic variants linked to a certain syndrome, which requires detecting more than a million single nucleotide polymorphisms (SNPs). This method empowers the researcher to identify not only new SNP in already known genes, but also completely novel genetic links.

## 3. Mendelian (Monogenic) Inheritance of ICH

Monogenic, also referred to as familial disorders, are rare genetic mutations following a clear Mendelian pattern of inheritance. They cause only a small number of ICH cases and tend to manifest at a younger age [11].

### 3.1. Familial Cerebral Amyloid Angiopathy (CAA)

Cerebral amyloid angiopathy is characterized by the depositions of amyloid-β peptide (Aβ), which is a cleavage product of the amyloid precursor protein (APP), in leptomeningeal arteries and cortical capillaries in the brain. Although numerous hypotheses have been proposed, the exact etiology of CAA remains unknown. It is unclear if the role of Aβ in the pathogenesis of the disease is causative, although the accumulation of Aβ fibrils in the vessels of the brain certainly correlates with the symptom occurrence and progression [12]. Typical manifestations include cognitive decline and intracranial hemorrhages [13]. Approximately 20% of all ICH are caused by CAA, affecting mostly the elderly population [13]. CAA-related hemorrhages are usually lobar and typically posteriorly localized [12]. The recurrence of the CAA-caused ICH is significantly higher, causing 7.4% annual recurrence risk in CAA-affected subjects, compared to 1.1% in the non-CAA group [14].

There are several genetic forms of CAA, sometimes classified as familial and sporadic. Familial types of CAA occur at a younger age and are characterized by a more severe course and a higher chance of an early lethal outcome [15]. Compared with the non-familial form of ICH, this CAA type is inherited in the Mendelian, autosomal dominant pattern [12]. The first detected gene related to familial CAA was the APP coding gene, located in the 21st chromosome [16]. SNPs in this gene cause early onset Alzheimer’s disease or CAA; whereas the variants affecting other Aβ regions manifest with prominent CAA. Polymorphisms in presenilin, cystatine C, and British precursor protein genes are associated with rare familial disorder forms characterized by ICH and the Alzheimer disease phenotype [12].

Sporadic CAA risk is caused by polygenic factors. This syndrome is described further in this article (part 4.3, “Lipid metabolism disorders”).

### 3.2. COL4A1 and COL4A2-Related ICH

Another frequent form of monogenic ICH is the *COL4A1*-related disorder. Similarly to CAA, *COL4A1* has familial and sporadic forms. Familial cases represent the rare autosomal dominant mutation of the α-1 chain of the type IV collagen encoding gene. This subtype of collagen plays a crucial role in the foundation and strengthening of basement membranes of the blood vessels and many other tissues [17]. The exact pathological mechanism is the inhibition of the heterotrimer deposition in the vascular basement membrane, due to missense mutations of highly conserved hydrophobic glycine residues, which results in the alteration of vessels structural properties [18].

Clinical manifestation covers different age periods as well as intra- and extracranial symptoms. Perinatal ICH and porencephaly, ICH of various localizations in those of adult age, microbleeds, lacunar ischemic strokes due to small-vessel disease, and leukoaraiosis are typical findings [19]. *COL4A1* mutations are also associated with cataracts, microcornea, Axenfeld–Rieger anomalies, and retinal hemorrhages [20]. Similarly, *COL4A2* mutations have been detected in ICH and porencephaly, suggesting the same pathological mechanism [21,22,23].

### 3.3. CADASIL

Cerebral autosomal dominant arteriopathy with subcortical infarcts and leukoencephalopathy (CADASIL) is caused by an autosomal dominant inherited mutation in the *NOTCH3* gene, and is characterized by alterations in arterial development and compromised cerebral hemodynamics [24]. CADASIL is well known as the most common monogenic cause of ischemic stroke in adults. The most common clinical findings include migraine with aura, ischemic events, mood disturbances, and progressing dementia [25]. However, there are also several documented CADASIL-associated cases of ICH [26]. ICH is an uncommon, yet possible, manifestation of CADASIL, occurring mostly in deep white matter of the brain in male adults. Most of the described patients with CADASIL-associated ICH had concomitant hypertension [27].

## 4. Polygenic Inheritance of the ICH Risk Factors

### 4.1. Hypertension

The most important risk factor for ICH is undoubtedly hypertension, which is an especially significant pathogenetic mechanism of deep white matter ICH [28]. This opinion is supported by the data from the International Stroke Genetics Consortium that found that the association of certain genes was linked to hypertension and intracerebral hemorrhage [29]. Despite the fact that no single SNP was proven to be specifically associated with ICH, the authors had constructed a genetic risk score (GRS) including 38 single nucleotide polymorphisms (SNP) known to increase the risk of hypertension. The analysis highlighted the association between the aforementioned GRS and deep ICH. Interestingly, this association was even more prominent in non-hypertensive subjects, whereas incorporating clinical hypertension into a model did not affect the risk substantially.

The most studied gene related to hypertension is the angiotensin-converting enzyme gene (*ACE*) located in the 17 chromosome. This gene participates in the regulation of the renin–angiotensin–aldosterone system (RAAS), thus, it is crucial for blood pressure and water–electrolytes balance. The main risk variant *rs1799752* has an insertion (I)/deletion (D) sequence in intron 16, which affects its activity and risk for stroke [30]. D allele was identified as a risk factor for ICH in Asian, but not Caucasian, populations [31,32,33].

### 4.2. Coagulation Disorders

Coagulation is a highly balanced system. Any dysregulation in it may lead to minor or major bleeding, including in the brain area. Some of the genetic polymorphisms associated with the deficits in the coagulation cascade are linked with ICH. Common variants of Factor XIII Subunit A (Tyr204/Phe204 and Leu564/Leu564) are associated with an increased risk of ICH in young white women (age < 45), especially in the presence of the plasminogen activator inhibitor polymorphism 5G/5G [34]. The -323Ins allele of Factor VII is associated with a 1.54-fold risk for ICH, whereas the -401G > T polymorphism of the same factor has no significant impact on hemorrhagic stroke [35,36]. As for factor V Leiden, the carriers seemed to be at lower risk of developing ICH [35]. Platelet-activating factor deficiency, caused by a missense V279F mutation in Exon 9, is also a genetic risk factor for ICH [37]. The β1-tubulin gene (*TUBB1*) Q43P polymorphism significantly increased both subarachnoid hemorrhage and ICH risk in men, and was associated with an earlier age of ICH occurrence [38].

### 4.3. Lipid Metabolism Disorders

The most studied and best-known ICH-causing genetic variant related to lipid metabolism is apolipoprotein E (APOE). *APOE* gene is located in the 19 chromosome and can be represented by three allele variants—ε2, ε3, and ε4. *Ε3ε3* is the most common genotype that is not related to ICH, whereas both ε2 and ε4 are missense variants involved in the pathogenesis of sporadic CAA [39,40], increasing the risk of subsequent lobar ICH [41]. A meta-analysis of 11 case–control studies showed a significantly increased frequency of the *APOE* ε4 allele in ICH cases in Caucasians and Asians [42]. The association of the ε4 with CAA is dose-dependent, whereas the ε2 allele is associated with larger hemorrhage volume that results in more severe disability and higher mortality [41,43]. It has been shown that the presence of the *APOE* ε2 and ε4 alleles puts the subjects on warfarin at substantial risk of lobar ICH: each ε2 and ε4 allele is associated with a 2.5-fold and 3-fold increase in the risk, respectively [44]. Despite the fact that earlier data clearly showed *APOE* association with lobar ICH, an increasing number of studies has proven both the ε2 and ε4 link to deep ICH, as well [42,43,45,46]. One of the studies has identified both ε2 and ε4 as increasing the risk for lobar ICH and only ε4 for the deep one [46].

*APOE* is not the only lipoprotein-coding gene that influences ICH risk. Apolipoprotein H (APOH) participates in lipid metabolism, hemostasis, and synthesis of the antiphospholipid antibody [47]. Four polymorphisms of the *APOH* gene were examined in the Chinese population. One of them (A allele of G341A) was observed significantly more frequently in ICH patients, especially those with a history of hypertension or family strokes [48].

Apolipoprotein (a) forms lipoprotein Lp(a) and is controlled by *LPA* gene expression. Its main risk variant is the repetitive pentanucleotide sequence TTTTA in the upstream of the transcription start site. A low number of TTTTA is related to elevation in the levels of Lp(a) and an increased risk of ICH [49].

Both Endoplasmic Reticulum Lipid Raft Associated 1 (*ERLIN1*) and Low-Density Lipoprotein Receptor (*LDLR*), located in 10 and 19 chromosomes, respectively, are important in cholesterol homeostasis. The main variants of those genes (rs1324694 for *ERLIN1* and rs688 for *LDLR*) are associated with a decreased risk of ICH [50,51]. Moreover, they are associated with a protective effect after the ICH.

Higher blood concentration of high-density lipoprotein cholesterol (HDL-C) has been described as a risk factor for ICH [52]. Another study found that genetic variants of the cholesteryl ester transfer protein gene (*CETP*) associated with increased HDL-C plasma levels also elevate risk of ICH with the strongest association with the *rs173539* SNP [53]. CETP inhibitors leading to an increase in the HDL-C levels in blood plasma are thought to be one of the potential atherosclerosis treatment options [54,55]. Unfortunately, the data about intracerebral hemorrhage in CETP inhibitors’ clinical trials are quite scarce so far [55]. It seems reasonable to investigate cerebrovascular outcomes in these trials further, including not only ischemic cardiovascular events, but also intracerebral hemorrhage frequency.

### 4.4. Inflammation

Inflammation plays an important role in vascular damage and the severity of stroke outcome. The upregulation of genes influencing inflammation processes could be associated with the increased risk of ICH. The methylenetetrahydrofolate reductase (MTHFR) coding gene is located in the 1p36.22 chromosome. The function of MTHFR is to convert amino acids’ homocysteine to methionine. The *rs1801133* polymorphism is the main risk variant, which reduces MTHFR activity and causes hyperhomocysteinemia, a well-known risk factor for inflammation, atherosclerosis, and endothelial dysfunction [56,57]. Large meta-analysis showed associations between ICH and *MTHFR* 677 T variant allele in both Asian and Caucasian populations [58].

Another essential cytokine in the inflammatory response to ICH is Interleukin 6 (IL6) [59]. It is responsible for endothelial cell activation and vascular dysfunction, as well as oxidative stress and an increased expression of pro-inflammatory cytokines. The rs1800796 polymorphism (C allele) is the functional promoter of IL6 and was noticed to be the major allele in the Asian population (79%) but the minor allele in the European population (<5%), according to the HapMap database. This genetic variant is related to an increased expression of IL6 and was significantly linked with an ICH risk in the Japanese population [60,61].

Another proinflammatory cytokine, initiating and regulating the inflammatory response, is tumor necrosis factor-*a* (TNF-*a*). The TNF-*a* concentration depends on different polymorphisms in the regulatory region. *rs1799964*, *rs1800629*, and *rs1800630* SNP are associated with increased TNF-*a* expression [62,63]. A study of the Taiwanese population showed that *rs1799964* and *rs1800629* are risk factors for deep ICH in men, while *rs1800630* has protective role [64].

The protein encoded by trafficking protein particle complex 9 (TRAPPC) is involved in the neuronal signaling of transcription factors and its overexpression enhances a cytokine-induced NF- κB signaling pathway [65]. An intronic polymorphism *rs12679196* (C allele) has been proved to significantly elevate the risk of ICH in the Japanese population, while the T allele decreased it [50].

Interferon Epsilon (IFNE) is expressed in several tissues, including brain, and is responsible for controlling immune reactions in the central nervous system [66,67]. The *rs2039381* polymorphism of the *IFNE* gene was an associated risk factor for ICH in Korean population [68].

Another study of the Korean population showed that the *rs2228048* polymorphism of transforming growth factor β2 receptor 2 gene *(TGFBR2)* increased the risk of ICH [69]. TGFBR2 is mostly expressed in neurons and plays an important role in T cells’ development and homeostasis [70,71].

Tissue inhibitors of metalloproteinases (TIMPs) are inhibitors for matrix metalloproteinases (MMPs)—endopeptidases, involved in blood–brain barrier breakdown and inflammatory reactions [72]. MMPs, especially gelatin-binding, are able to degrade type IV and V collagen thus break down the vascular basement membrane and cause the rupture of the vessel walls [73,74]. *TIMP-1* and *TIMP-2* are located in the X and 13 chromosomes, respectively. The main *TIMP-1* risk variant for ICH is *rs2070584*, whereas another variant (*rs4898)* provides a protective effect on deep ICH [75,76]. As for *TIMP-2*, the main risk variants are *rs7503726* and *s7503607*; however, *rs7503726* also was considered as protection against deep ICH in some populations [77,78].

Moreover, genes coding MMP themselves might also contribute to the ICH risk. SNPs in the MMP2 (*rs2285053*) and *MMP9* (*rs3787268*, *rs2250889*) genes increase the risk of deep ICH in the older Asian population (>65 years old). The pathogenetic mechanism is thought to be extracellular matrix degradation promoted by MMP [79].

### 4.5. Small Vessel Disease

A recent study of genetic loci associated with stroke has found eight genes related to ICH. Four loci (*EDNRA, 1q22, MMP12, SH3PXD2A*) were the same in ICH and small vessel stroke, suggesting a mutual underlying mechanism [80]. An earlier study revealed *1q22* as a susceptibility locus for deep ICH with the main risk variant *rs2984613*. In the case of this SNP, each additional C allele increased the frequency of deep ICH by 24% [81].

### 4.6. Others

Meta-analysis by Woo et al. revealed new susceptibility loci for ICH. Polymorphisms in the *12q21.1* region (especially the *rs11179580* variant) were associated with lobar ICH, but only in the European population. The exact mechanisms of the ICH caused by this gene polymorphism are quite unclear [81]. A large study by Traylor et al. has found eight genes related to ICH. Seven of them (*1q22, COL4A2, EDNRA, LINC01492, MMP12, SH3PXD2A, CDK6*) were influencing both ischemic stroke and ICH, while *ANK2* (*rs149538932*) raised risk predominantly in ICH. As mentioned before, four of them (*EDNRA, 1q22, MMP12, SH3PXD2A*) are related to small vessel disease, whereas the exact mechanism of other polymorphisms is still unknown [80]. The summary of the most well-known genes and SNPs associated with the increase in polygenic ICH risk is provided in Table A1 (Appendix A).

## 5. Genetic Predictors of Outcome

Genetic factors could provide a hint regarding ICH prognosis. Newly discovered susceptibility risk locus in the *17p12* is associated with hematoma volume, clinical severity, and functional outcome in non-lobar ICH in patients of European ancestry [82]. The *APOE* ε2 allele is associated with a larger ICH volume, as well as an elevated risk of mortality and poor functional recovery in lobar ICH [83]. Inflammation also has an impact on ICH severity and outcome: high IL-6 plasma levels at admission could predict hematoma enlargement [84].

## 6. Gene Therapy Opportunities

So far there is no effective, specific treatment for ICH, although recent studies suggest promising results. The greatest interest is raised by strategies targeting secondary brain injury, in particular, mechanisms such as inflammation, oxidative stress, apoptosis, brain edema, and neuronal damage [85]. Therefore, genetic studies are crucial, as they provide information on mechanisms involved in the ICH pathogenesis and provide more possibilities for development of target-specific treatment.

Recent studies revealed some important interactions between the inflammatory gene expression in peripheral blood and ICH. One of them identified 122 genes with altered expression in ICH patients, mostly involved in neuroinflammation, apoptosis, interleukin, or transcriptional regulation. It also showed significant overlap between dysregulated genes in blood and perihematomal brain tissue, suggesting the importance of blood–brain interactions in ICH patients [86].

Another study performed an analysis of expressed genes in peripheral blood within 24 h of ICH onset and 72 h after the first sample. According to the results, the expression of 218 genes was significantly altered after ICH; the most-induced genes were related to the inflammation and the activation of the immune response [87]. Both of the studies open paths for novel target-specific treatment investigations, although they also require additional research on links between the study subject and ICH pathogenesis, and outcomes.

## 7. Conclusions

ICH is a severe disease resulting in high mortality and a large proportion of unfavorable functional outcomes, sometimes being recurrent and causing even more devastating consequences. Such a disorder requires timely risk factor evaluation and prevention. Genetic risk factors play an important role in the ICH pathogenesis, presenting by a large spectrum of pathologies, including hypertension, neurovascular inflammation, disorders of lipid metabolism, and coagulation cascade. Genetic data should be used for the analysis of potential ICH mechanisms and to optimize the prevention: e.g., patients who are genetically susceptible to deep ICH, which is frequently caused by hypertension, possibly should receive more strict blood pressure control. This information would enable clinicians to individualize ICH prevention and treatment strategies, and to plan the outcomes. So far, existing data seems largely of theoretical and scientific value; nevertheless, experimental research in genetics is rapidly evolving, giving us more hope regarding ICH patients’ management in future.

## Data Availability

Not applicable.

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
