# Peer review of "Intracerebral Hemorrhage Genetics"

_genes, 2022, doi:10.3390/genes13071250_

Round 1

Reviewer 1 Report

Ekkert et al. reviewed the literature, and summarized in the manuscript the studies on intracerebral hemorrhage (ICH) genetics. My primary concern is that a recently published review paper (Front Neurosci. 2022;16:874962) covers most of what are described in the current manuscript that I found provides little new knowledge in the area.

Specific comments   

1) The authors state in the Abstract that “In this paper we describe gene-related ICH mechanisms, discuss the emerging treatment options and possible clinical relevance of the genetic findings in ICH management”. However, in the text, there is limited description of the mechanisms, and scarce discussion on the emerging treatment options possibly due to the lack of treatment options derived from the genetic findings.

2) The authors write in lines 88-89 that “Sporadic CAA risk is caused by polygenic factors. This syndrome is described further in this article”, but where is the further description of the syndrome?

3) A full picture of key information from a study reviewed should be provided. Please add the following points in the text: no single SNP was found to be associated with ICH in the International Stroke Genetics Consortium study (for lines 123-127); subjects with factor V Leiden had decreased risk for ICH (for ref 35, line 142); etc. 

4) Ref 37 in the reference list does not exist.

5) Minor grammatical errors: change “The coagulation is highly balanced system” in line 135 to “The coagulation is a highly balanced system”; change “-323Ins allele” in line 140 to “The -323Ins allele”; etc.

Author Response

1) The authors state in the Abstract that “In this paper we describe gene-related ICH mechanisms, discuss the emerging treatment options and possible clinical relevance of the genetic findings in ICH management”. However, in the text, there is limited description of the mechanisms, and scarce discussion on the emerging treatment options possibly due to the lack of treatment options derived from the genetic findings.

Corrected --> "In this paper we describe gene-related ICH types and some of underlying mechanisms. We also briefly discuss the emerging treatment options and possible clinical relevance of the genetic findings in ICH management."

2) The authors write in lines 88-89 that “Sporadic CAA risk is caused by polygenic factors. This syndrome is described further in this article”, but where is the further description of the syndrome?

It is described in part 4.3 (Lipid metabolism disorders), but to avoid misunderstanding we have corrected: --> "Sporadic CAA risk is caused by polygenic factors. This syndrome is described further in this article (part 4.3., "Lipid metabolism disorders").

3) A full picture of key information from a study reviewed should be provided. Please add the following points in the text: no single SNP was found to be associated with ICH in the International Stroke Genetics Consortium study (for lines 123-127); subjects with factor V Leiden had decreased risk for ICH (for ref 35, line 142); etc. 

Corrected as requested.

4) Ref 37 in the reference list does not exist.

Corrected (https://www.ahajournals.org/doi/10.1161/01.str.28.12.2417).

5) Minor grammatical errors: change “The coagulation is highly balanced system” in line 135 to “The coagulation is a highly balanced system”; change “-323Ins allele” in line 140 to “The -323Ins allele”; etc.

Corrected as requested.

Reviewer 2 Report

This is a review article addressing the current knowledge on monogenic and polygenic causes of intracerebral hemorrhage (ICH) genetics. The authors attempt to discuss ICH-related mechanisms, emerging treatment options and the significance of genetic findings in ICH.

The authors initiate their review by describing what ICH is and the impact on society. They then go on to describe the major monogenic causes of ICH- familial cerebral amyloid angiopathy; COL4A1 and COL4A2-related ICH and cerebral autosomal dominant arteriopathy with subcortical infarcts and leukoencephalopathy (CADASIL) and polygenic causes: hypertention; coagulation disorders; lipid metabolism disorders; inflammation, small vessel disease and finally they mention additional factors identified in ICH.

Overall this is a coherent, understandable and an easily read review with a clear message. Ideally, it should have been accompanied by a graphic showing familial and polygenic causes of ICH to make it more digestable. Apart for its minor grammatical errors my major comments are the following.

1. Line 145: SAH is not defined.

2. Lines 180-182: It is not clear what the authors are trying to convey in this sentence- what are the CETP inhibitors used for and how exactly do they affect HDL-C? Please clarify.

3. There is no mention of the Table in the main text.

Minor grammatical comments

1. L27: It is AN important...

2. L47: ICH, but HAS little....

3. L58:more than A million

4. L73: mostly THE elderly

5. L82: was THE APP

6. L92-93: represent THE rare...of THE type IV

7. L96-97: inhibition of THE heterotrimer...glycine residueS

8.L119: which is AN especially...

9.L135-136: coagulation is A highly...including ones in THE brain area

10. L139: in THE presence

11.L151: missence variants are involved

12. L180: in THE rs173539

13. L194: Interleucin 6

14. Please use either IL6 or IL-6 not both.

15.L196-243: Please go through this part of the text and correct with nouns (eg. a) and articles (eg. the)

Author Response

1. Line 145: SAH is not defined.

Corrected.

2. Lines 180-182: It is not clear what the authors are trying to convey in this sentence- what are the CETP inhibitors used for and how exactly do they affect HDL-C? Please clarify.

Corrected (citations provided in the manuscript) --> Higher blood concetration of high-density lipoprotein cholesterol (HDL-C) has been described as a risk factor for ICH [52]. Another study found that genetic variants of cholesteryl ester transfer protein (CETP) associated with increased HDL-C plasma levels also elevate risk of ICH. The strongest association has been found in rs173539 locus [53]. CETP inhibitors leading to an increase of the HDL-C levels in blood plasma are thought to be one of the potential atherosclerosis treatment options. Unfortunately, the data about intracerebral hemorrhage in CETP inhibitors’ clinical trials are quite scarce so far. It seems reasonable to investigate cerebrovascular outcomes in these trials further including not only ischemic cardiovascular events, but also intracerebral hemorrhage frequency.

3. There is no mention of the Table in the main text.

Corrected.

All the minor grammatical errors have been corrected accordingly to the reviewer's comments.

Round 2

Reviewer 1 Report

I do not think the manuscript has improved significantly after revision.

A more comprehensive review on the genetics of ICH has recently been published (Front Neurosci. 2022;16:874962), and the current manuscript provides little new knowledge. This is my primary concern raised in the last round of review, which remains unaddressed.